# Lee waves detection over the Mediterranean Sea using the Advanced Infra-Red WAter Vapour Estimator (AIRWAVE) Total Column Water Vapor (TCWV) dataset

Enzo Papandrea[1,2], Stefano Casadio[1,3], Elisa Castelli[2], Bianca Maria Dinelli[2], and Mario Marcello Miglietta[2]

[1]Serco Italia s.p.a., via Sciadonna 24-26, 00044 Frascati, Italy
[2]Istituto di Scienze dell'Atmosfera e del Clima, ISAC-CNR, Via Gobetti 101, 40129 Bologna, Italy
[3]European Space Agency (ESA/ESRIN), Via Galileo Galilei 1, 00044 Frascati, Italy

**Correspondence:** Enzo Papandrea (e.papandrea@isac.cnr.it)

**Abstract.** Atmospheric gravity waves generated downstream by orography in a stratified airflow are known as lee waves. In the present study, such mesoscale patterns have been detected, over water and in clear sky conditions, using the Advanced Infra-Red WAter Vapour Estimator (AIRWAVE) Total Column Water Vapour (TCWV) dataset, which contains about 20-years of day and night products, obtained from the thermal infra-red measurements of the Along Track Scanning Radiometer (ATSR) instrument series. The high accuracy of such data, along with the native $1\times1$ km$^2$ spatial resolution, allows the investigation of small scale features such as lee waves. In this work, we focused on the Mediterranean region, the largest semi-enclosed basin on the Earth. The peculiarities of this area, which is characterized by complex orography and rough coastlines, lead to the development of these structures over both land and sea. We developed an automatic tool for the rapid detection of areas with high probability of lee waves occurrence, exploiting the TCWV variability in spatial regions with $0.15°\times0.15°$ extent. Through this analysis, several occurrences of structures connected with lee waves have been observed. The waves are detected in spring, autumn and summer seasons, with TCWV values usually falling in the range from 15 to 35 kg m$^{-2}$. In this article, we describe some cases over the Central (Italy) and the Eastern Mediterranean basin (Greece, Turkey, Cyprus). We compared a case of perturbed AIRWAVE TCWV fields due to lee waves occurring over the Tyrrhenian Sea on 18 July 1997 with the sea surface winds from the Synthetic Aperture Radar (SAR), which sounded the same geographical area, finding a good agreement. Another case has been investigated in detail: on 2 August 2002 the Aegean sea region was almost simultaneously sounded by both ATSR-2 and AATSR instruments. The AIRWAVE TCWV fields derived from the two sensors were successfully compared with the vertically integrated water vapour content simulated with the Weather Research and Forecasting (WRF) numerical model for the same time period, confirming our findings. Wave parameters such as amplitude, wavelength and phase, are described through the use of the "Morlet" Continuous Wavelet Transformation (CWT). The performed analysis derived typical wavelengths from 6 to 8 km and amplitude up to 20 kg m$^{-2}$.

# 1 Introduction

Atmospheric gravity waves (AGW) may be generated in a stably stratified atmosphere when an airflow impinges on orographic obstacles, like isolated mountains or islands. AGW may develop in both upstream and downstream directions. However, in situations where the wave energy is ducted close to the surface, the motion is confined downstream in the lower atmosphere and the waves are located within a wedge-shaped wake behind the obstacle (Vosper et al., 2006). Most of the observed AGW fall in the latter group and they are referred to as trapped "lee waves" (Li et al., 2011). The waves may be classified into two different categories: a) diverging wave type, characterised by crests orientated outwards from the centre of the wake; b) transverse wave type, where the crests are nearly perpendicular to the wind direction (Gjevik and Marthinsen, 1978). Lee waves may play an important role in modifying the vertical structure of wind, moisture, and temperature near and behind an obstacle, being nowadays a well-recognized hazard to aviation, especially under dry conditions, when they do not generate clouds and, thus, cannot be easily detected (Christie, 1983; Uhlenbrock et al., 2007). In specific circumstances (e.g. as a consequence of the interaction with extensive mountain ranges), non-linear perturbations may also interact with larger-scale flow, impacting on the global scale circulation and the climatological momentum balance (Scorer, 1949; Wurtele et al., 1993; Teixeira et al., 2013; Teixeira and Miranda, 2017; Teixeira et al., 2017). Therefore, the study of such phenomena is of broad relevance. Lee waves can be identified through the use of satellite instruments, e.g. the Synthetic Aperture Radar (SAR), that can detect, in the backscattered radar power, the small-scale sea surface roughness, enabling to estimate the varying wind speeds at the sea surface (Cheng and Alpers, 2010).

Background concentrations of atmospheric constituents, such as the Total Column of Water Vapour (TCWV), are also perturbed by AGW. Therefore, TCWV fields, derived e.g. from satellite instruments, can also be used for lee waves identification. The only constraint is that their accuracy and resolution have to be sufficiently high. Water vapour is a highly variable gas, its concentration can span several orders of magnitude in the lower atmosphere. It has a major role in transporting latent heat and, for this reason, is a principal element in the thermodynamics of the atmosphere (Jacob, 2001). It is the most important natural greenhouse gas in the Earth's atmosphere, accounting for about half of the present-day greenhouse effect, and it is the dominant gaseous source of infra-red opacity in the atmosphere (Held and Soden, 2000). Accurate knowledge of atmospheric water vapour is therefore important and a big effort is made by Space Agencies and by the Scientific Community to globally derive its abundance using a wide range of techniques and observational platforms. Lee waves produce a deepening of the moist layer and oscillations in different atmospheric variables, including vertical velocity. As a consequence, the air is alternately lifted and lowered, causing convergence and divergence of air in the warmer moist bottom layer, which is seen as bands in the TCWV field (Lyapustin et al., 2014). The columnar water vapour fields obtained from the Moderate Resolution Imaging Spectroradiometer (MODIS) (Salomonson et al., 2002) has already been used to observe mountain lee waves over land: water vapour is derived from the 6.7 $\mu$m channel whose peak sensitivity is at 550 hPa (about 5 km) in the free troposphere (Uhlenbrock et al., 2007). More recently, mountain lee waves have also been identified in the MODIS TCWV product derived from Near-InfraRed (NIR) 0.94 $\mu$m measurements, which are sensitive to atmospheric features very close to the surface (Lyapustin et al., 2014). The perturbations of the vertical structure of the planetary boundary layer, causing quasi-periodic oscillations of

columnar water vapour, aerosol optical depth, and surface irradiance in different spectral bands have also been detected using ground-based observations collected at Lampedusa, in the central Mediterranean Sea (Di Sarra et al., 2013). The authors were also able to estimate the Saharan dust radiative forcing variations during the gravity wave event.

Miglietta et al. (2010) showed that the simulations performed with the Weather Research and Forecasting (WRF) model are able to reproduce the near-surface wind speed variability due to gravity waves. Model lee wave features are similar to the sea surface wind patterns extracted from the ENVIronmental SATellite (ENVISAT) Advanced Synthetic Aperture Radar (ASAR) images, proving that the model is able to represent such phenomena in a realistic way.

Recently, Casadio et al. (2016) have developed the Advanced Infra-Red WAter Vapour Estimator (AIRWAVE) algorithm to obtain TCWV from the measurements of the Along Track Scanning Radiometer (ATSR) instrument series. The application of the AIRWAVE algorithm (Casadio et al., 2016) to the measurements of the ATSR missions (Delderfield et al., 1986) has produced a dataset of about 20 years of day and night TCWV, retrieved over water in clear sky conditions, at the native $1 \times 1$ km$^2$ spatial resolution grid. The dataset has been recently used to detect trends in the Inter Tropical Convergence Zone (ITCZ) latitudinal displacement over the duration of the combined ATSR missions (Castelli et al., 2018a). The quality of the AIRWAVE products has been verified using both satellite and radiosonde correlative measurements (Papandrea et al., 2018; Schröder et al., 2018). To date, the dataset is available in two versions; the latest version (V2) is obtained with an improved version of the AIRWAVE algorithm which accounts for the atmospheric variability at different latitudes and associated seasonality, reducing both biases and root mean square error, especially in polar and coastal regions (Castelli et al., 2019). The overall good quality of the dataset suggested that it could be used for systematic lee wave detection. Indeed, oscillations of the TCWV in regions where lee waves can form have been found during quality checks of the AIRWAVE dataset (Castelli et al., 2018b).

In this article we describe a method developed for the detection of wave structures in the AIRWAVE dataset, focusing on the Mediterranean region. The detections are obtained through an automatic tool that exploits the TCWV variability in spatial regions of appropriate extension. A subset of the identified occurrences is compared both with correlative measurements from the SAR instrument onboard the European Remote Sensing (ERS)-2 satellite and with WRF model simulations, strengthening the reliability of the lee waves identification in the AIRWAVE dataset.

The paper is organized as follows: in section 2 we describe the approach adopted for the automatic lee wave detection and we show some selected occurrences; in section 3 we describe a case on 18 July 1997, comparing the AIRWAVE TCWV values with the SAR/ERS-2 fields, and a case on 2 August 2002 (when both ATSR-2 and AATSR sounded the Aegean sea region at very close times), comparing the AIRWAVE TCWV fields with WRF simulations; in section 4 we perform a wave characterization using the Continuous Wavelet Transformation (CWT) and, finally, conclusions are given in section 5.

## 2   Lee waves detection using the AIRWAVE dataset

The quality of the AIRWAVEv2 TCWV dataset (Castelli et al., 2019) was verified not only using external correlative measurements but also adopting internal quality checks (e.g. range of validity of the TCWV values and their spatial variability). The analysis highlighted the overall good quality of the dataset but also showed a recurrent higher variability in specific areas in

the Mediterranean region. We noticed that all these occurrences were associated with lee wave patterns, generated by airflows crossing one or more obstacles (Castelli et al., 2018b). These phenomena indeed induce water vapour perturbations very close to the surface, producing strong variations in TCWV. Considering the small scale of these patterns, the native 1x1 km$^2$ spatial resolution of the AIRWAVE dataset is suitable to capture the TCWV variations associated with lee waves.

Given that AIRWAVE TCWV could detect lee waves, we designed a systematic approach for the statistical analysis of these events. The very large number of ATSR products, covering about 20 years of measurements, requires an automatic and efficient tool for a systematic detection of the waves. The basic idea for the development of this tool is that the presence of waves produces a local enhancement in the TCWV variability. To detect this variability enhancement, we divided the examined region into regular grid cells of dimensions fine enough to capture the geographic details of the studied areas but

large enough to contain an appropriate number of measurements to reliably determine the variability. For each grid cell, we computed the standard deviation (STD) of the observed TCWV. Background STD was estimated as the standard deviation of the monthly mean computed in grid cells in Mediterranean areas where lee waves occurrence was unlikely. For this purpose, the identified areas have been selected on the bases of wind speed and direction ($< 2$ m s$^{-1}$ and not downwind to land). We then evaluated the Normalised STandard Deviation (NSTD) as the STD found in the examined areas divided by the background

STD. The normalization was performed in order to minimise the dependence of the results on intrinsic atmospheric variability and instrument characteristics (e.g. measurements noise) and allows to use the same classification criteria for all the instruments in the ATSR series (ATSR-1, ATSR-2 and AATSR) and for all the seasons. Based on ERA-Interim reanalysis, Wypych et al. (2018) provided an analysis of temporal and spatial variability of TCWV for Europe. Differences strongly depend on air temperature and on latitude; other determinants include local factors, such as the presence of water or land. Atmospheric

circulation is a key factor for the moisture content in winter. In contrast, evaporation from the sea provides a relevant source of moisture in Mediterranean areas especially in autumn, when the air temperature is still high and the air is able to absorb water vapor emitted by the heated sea surface. The interannual variability of TCWV over the Mediterranean Sea is small, with changes less than 4.5 kg m$^{-2}$ and a fairly even distribution. The seasonal variability is also relatively small in the Mediterranean, being less than 20 % of the monthly mean TCWV in summer. For these reasons, the influence of the variation of the baseline TCWV

on the value of NTSD is limited for our purposes.

  The size of the grid cells is a critical issue in our method: different grid sizes have been tested and a regular latitude-longitude grid, with each grid cell covering an area of $0.15° \times 0.15°$ (about $17 \times 13/14$ km$^2$ at 35/40 latitude degrees) was found to give the best performance in terms of both the ability to capture coastline details and the robustness of statistics. Each cell is included in our analysis only if the fraction of cloud-free TCWV products is larger than 20 % (in order to ensure robust statistics).

At latitudes typical of the investigated areas, these choices mean that the random uncertainty (noise) in the mean TCWV is reduced, by averaging within each grid cell, to less than 1 % (the single pixel noise error on TCWV was estimated to be 4.8 % / 5.2 % for AATSR / ATSR-2 at mid-latitude, for more details see Castelli et al. (2019)).

  The scenes have been classified using two criteria, both applied on the latitude-longitude regular grid. The first is based on the percentage of cloud-free grid cells (we label the associated threshold as TR1) and the second on the percentage of grid cells

having NSTD values exceeding the value of 2 (labeled as TR2). The aim of this selection scheme is to detect the most evident

cases of lee waves in scenes characterized by a sufficient number of available (i.e. cloud-free) measurements. The classification of the most representative events was populated with the scenes where both the criteria TR1 > 70 % and TR2 > 1 % were satisfied. We mainly considered, for the lee waves detection, three Mediterranean sub-basins, which are shown in the top-left map of Fig. 1. The first two are located over the central Mediterranean Sea and include mainly the Tyrrhenian Sea (R1) and

the Southern Tyrrhenian Sea / Ionian Sea (R2), while the other is located on the Eastern Mediterranean Sea (R3). In the whole AIRWAVE dataset, we found about thirty occurrences in R1+R2 and fifty in R3. The relatively low number of detections is related to the revisit time over one specific location (about 3 days) due to the limited ATSR across track swath (about 500 km wide).

Figure 1 shows the total number of monthly TCWV wave occurrences as a function of the month of the year (black curve).

About 76 % of the selected cases are detected in JJA (June-July-August), 17 % in MAM (March-April-May) and the remaining 7 % scenes in SON (September-October-November). Since we use only "cloud-free" measurements to detect the TCWV wave occurrences, the number of identified cases depends upon the amount of available cloud-free data. In order to estimate the typical number of cloud-free pixels in each month, we performed an analysis using all the available AATSR measurements for the whole year 2003. In Fig. 1 the monthly means of the percentage of cloud-free measurements are also plotted, using different

colours according to the different geographical regions shown on the map. The highest number of available observations is found in August, in particular over the Eastern Mediterranean Sea (about 80 %), while the lowest observations amount is found in the winter (December-January-February, DJF) (about 20 %), correspondingly closely to the number of wave detections. One reason for no TCWV wave occurrence in the winter months could be attributed to the frequent presence of lee waves accompanied by cloud.

The Aegean Sea is particularly suitable for the observation of these phenomena. From spring to fall, the Northern Aegean is often crossed by north-easterly winds, called Etesians (Kotroni et al., 2001; Miglietta et al., 2017), that change direction (becoming north-westerly or westerly) and intensify over the Southern Aegean, approaching JJA monthly averages of about 8-10 m s$^{-1}$ in part of the Cyclades, of the Dodecanese Islands and East of Crete (e.g. Vagenas et al., 2017). Anticyclonic circulations, typical of summertime in the region, often yield clear sky conditions. The complexity of the coastal orography

and the presence of mountainous islands deeply influence the local-scale atmospheric circulation in the Ekman layer, producing effects at spatial scales down to a few kilometres (Vagenas et al., 2017).

As an example of TCWV wave detection, Fig. 2 shows the AATSR TCWV fields (left panel) and the corresponding NSTD over the Aegean Sea (right panel) for 2 August 2002. In this figure, the wind speeds and directions, obtained from ERA-Interim products at grid resolution 1.0°×1.0°, are indicated with white arrows. The reported speeds have been derived using

a linear interpolation over time of the 10 m (u and v) wind components, available every six hours. The left panel of Fig. 2 clearly shows the lee wave structure of the ATSR TCWV, while the right panel of the same figure, showing the NSTD, clearly demonstrated the correct geolocation of the lee wave area derived using our tool. In correspondence with the TCWV waves, the NSTD value increases with respect to the background values, showing patterns clearly correlated with the wind direction. The used approach, which makes use for simplicity of a regular aggregation box size used to calculate the NSTD, works almost

uniformly for the majority of typical wavelengths. Nevertheless, small effects may arise both in case of wavelengths lower than

half of the box size (wave effects are not completely resolved) or in case of wavelengths approaching the box size (only part of the wave is captured within the box), resulting in a decreasing of the NSTD value.

Several lee wave occurrences have been found close to the island of Crete, which has an elongated shape, spanning 260 km from East to West, while its North-South extension ranges from 12 to 60 km. The shape of the island and the presence of elevated mountain regions (up to 2450 m a.s.l.) generate wave-like patterns. Other occurrences have been found close to Sardinia and Corsica and near Cyprus.

Figure 3 shows the derived NSTD for a set of selected lee wave events. Two cases are located between Sardinia, Corsica and the Italian Peninsula, over the Tyrrhenian Sea (panels g and h). One event is located in the Eastern Mediterranean across the island of Cyprus (panels f) and the remaining five over the Aegean Sea (panels a-e). In the figure, two cases are measured at two different times of the same day: panels a) and b) are obtained from, respectively, ERS-2 morning and evening satellite passes for the same day with local crossing time at approximately 10:20 (descending) and 22:20 (ascending).

The ECMWF wind speeds for the selected events are shown with white arrows superimposed to the mapped NSTD. The figure indicates that the patterns of the normalised standard deviation are evidently correlated with the wind direction. The atmospheric conditions of the reported cases, all characterized by intense and constant wind direction, allowed the AGW propagation over long horizontal distances, trapping the wave vertically. Wind jets in the lee of mountain ranges may indeed generate horizontal waves which may propagate in a wave duct or wave guide horizontally over distances even of several hundreds of kilometres (Cheng and Alpers, 2010).

## 3 Validation of the Lee waves detections

In order to find scientific evidence that the detected structures are indeed lee waves, we selected a subset of lee wave occurrences and we compared them with both independent observations and model simulations. We found one particular case where the lee waves were measured almost simultaneously by two ATSR instruments. The comparison of the results of the detection algorithm applied to the two instruments enabled the assessment of its robustness, since each instrument has its own characteristics (e.g. noise, systematic errors), which may act in a different way on their measurements. This lee wave detection was then compared to the outputs of the WRF limited area numerical model (see Sect. 3.1).

We also searched for SAR sea surface wind data co-located in time and space with AIRWAVE TCWV wave detections. We found one particular case suitable for our purposes from the ERS-2 SAR (see Sect. 3.2).

In all the aforementioned comparison exercises, we observed the presence of similar structures both in the AIRWAVE dataset and in the correlative data, thus enhancing the level of confidence of the detected AIRWAVE lee wave events (see Sect. 3.1).

The performed comparisons are illustrated in more details in the following subsections.

### 3.1 ATSR-2 and AATSR near simultaneous measurements and WRF model comparison

On 2 August 2002 the Aegean Sea area was observed by both ATSR-2 (onboard ERS-2 satellite) and AATSR (on board ENVISAT) with almost the same orbit track and a time separation of around 30 minutes. This "tandem" configuration offers

an important opportunity to compare the TCWV products of the two sensors, which are similar in design but are characterised by different random and systematic measurement errors.

North-westerly winds were blowing with high intensity. Furthermore, the cloud masks of the two sensors indicate that the sky over the sea was almost entirely cloud-free. Our algorithm detected lee waves with similar spatial distributions from the TCWV fields derived from both sensors. In particular, the lee waves were located in the South-East of Central Greece (Evia Island) and of numerous Cyclades and Dodecanese Islands (Andros, Tinos, Amorgos, Ikaria, Samos, Kos, and, to the South, up to Crete, Karpathos and Kassos), as can be seen in the TCWV fields and in the corresponding NSTD map shown in Fig. 2.

We investigated in more detail the geographical area surrounded by the red box in Fig. 2. Figure 4 reports the AIRWAVE TCWV values of this region, for both ATSR-2 and AATSR, in the left and right panels respectively. In this region, the Amorgos island has an important role in the development of the lee waves. Amorgos is a long and narrow island, with a length of 32 km and a width ranging between 1.9 and 5.5 km. It has relatively high mountains (three of them over 500 m a.s.l.) that fall steeply into the sea. The wind direction on 2 August 2002 was almost perpendicular to the island main axis, as indicated by the ECMWF ERA-Interim wind speeds and directions, which are superimposed in the figure. These conditions led to the formation of observed trapped waves, that extended several kilometres away from the source, clearly visible in the figure as alternate darker and lighter stripes. In Fig. 4 a very large correlation between the two AIRWAVE datasets can be noticed. This suggests that the observed pattern is real and not an artefact of a particular sensor.

To strengthen the robustness of our analysis, we have used the WRF model to simulate the atmospheric conditions of the sounded area. The Advanced Research WRF model (WRF-ARW version 3.5.1; Wang et al. (2010)) was implemented with a grid spacing sufficiently fine for the simulation of small-scale atmospheric waves. In this work, it was implemented in a one-way nested 3-grid configuration, with 40 vertical levels and horizontal grid spacing respectively of 16, 4, and 1 km. $400 \times 400$ grid points are used in the inner domain, covering the area ($33.6\,^\circ$N – $37.4\,^\circ$N, $24.7\,^\circ$E – $29.2\,^\circ$E). ECMWF analysis/forecasts are used as initial/3-hourly boundary conditions. The model simulation starts at 00:00 UTC, 2 August 2002, and lasts for 24 hours, so that the times of satellite observations are definitely after the model spin-up time. The following parameterization schemes are employed: the Thompson et al. (2008) microphysics; the Rapid Radiative Transfer Model (RRTM) for longwave radiation (Mlawer et al., 1997); the Dudhia (1989) scheme for shortwave radiation; the unified Noah land-surface model (Niu et al., 2011); Mellor–Yamada–Janjić for the planetary boundary layer (Janjić, 2001). In the past, the model was successfully employed, with a similar configuration, to simulate wind speed in two case studies of orographic lee waves over the Eastern Mediterranean Sea in comparison with SAR retrieval (Miglietta et al., 2013).

In Fig. 4, the modeled vertically integrated water vapor values are superimposed to the ATSR TCWV fields (for the sake of image readability, only two contour levels are shown). The model foresees the presence of waves, with similar periodicity and direction as the ones detected by AIRWAVE, capturing very well the South-Eastward lee waves extending from the Cyclades islands of Amorgos and Anafi. The presence of a wave-like pattern variability in the lower troposphere was also confirmed from the WRF geopotential and temperature fields (not shown).

## 3.2 ATSR-2 and SAR comparison

An opportunity for comparison with independent correlative data was found for the event on 18 July 1997. During that day, the Tyrrhenian Sea region was observed by both the ATSR-2 instrument and the SAR instrument, both onboard ERS-2 satellite. Strong westerly winds blew across the island of Sardinia and in the Tyrrhenian Sea, with direction and strength almost constant over the considered basin.

An enhancement of the NSTD, derived from the ATSR-2 AIRWAVE TCWV fields of the ERS-2 orbit 11734, was detected in many grid cells of the considered region, as shown in the top panel of Fig. 5. The corresponding TCWV values, shown in the central panel of the figure, show that the winds crossing Sardinia (from the ERA-Interim fields at $1.0° \times 1.0°$) clearly generate wave patterns over the sea, induced by the orography of the island. The formed lee waves propagate downstream for hundreds of kilometres in the Tyrrhenian Sea towards the Italian peninsula. Once again, the NSTD enhancements are largely correlated with the lee wave patterns in the TCWV fields. However some NSTD values appear anomalously high, particular in areas where there is missing data due to cloud. It may be some thin cloud is missed in the ATSR mask, leading to error (increased variability) in the retrieved TCWV. A secondary cause may be the presence of clouds, reducing the number of elements that can be used within the grid cells. The observed TCWV fluctuations can be compared to the SAR wind fields, shown in the single look complex images for the same day reported in the bottom panel of Fig. 5. The picture has been obtained using the Earth Observation Link (EOLi, https://earth.esa.int/web/guest/eoli) European Space Agency's client for Earth Observation Catalogue Service. The EOLi tool allowed the selection of Earth Observation products acquired by the ERS and ENVISAT satellites and the display of the related images on the top of an orthographic representation of the Earth. The service has been recently replaced by the ESA Simple Online Catalogue (https://esar-ds.eo.esa.int/oads/access/).

The spatial resolution of the SAR/ERS-2 images is very high: 26 m in range (across track) and between 6 and 30 m in azimuth (along track). The swath width of the sensor is about 100 km, thus covering only a portion of the AIRWAVE products in the longitude domain. The lee wave patterns shown in the SAR images downstream of the island of Sardinia reflect the sea surface impression of the perturbations in the lower troposphere. The patterns are in agreement with those derived from the AIRWAVE dataset. In particular, the geographical regions where the trapped lee waves are located are consistent between the two sensors (red boxes are shown in the two panels as a guide).

## 4 Lee wave characterization

As described in Section 2, the first step of our methodology consists in the detection of lee waves occurrence. Once the lee waves are found, it is useful to have a tool to provide quantitative information about the "local" wave characteristics, such as amplitude and wavelength estimated in different locations within the considered area. The fact that lee waves are stationary waves enables wave parameters to be recovered from the 2-D spatial patterns observed by satellite.

Small variations in the lee wave induced features may occur, but their time scale is much longer than the duration of the satellite overpass, so that they should not affect our results. We adopted the wavelet approach proposed by Torrence and Compo (1998), which was found suitable to characterize signals. The method used here is based on the "Morlet" Continuous Wavelet

Transformation (CWT), a tool that can be used to analyze scale-dependent structures of a signal as it varies in time (or in space) (Sadowsky, 1996). Several practical aspects of the analysis are reported in Torrence and Compo (1998), using time series of the El Niño–Southern Oscillation (ENSO) as a possible example of its application. The CWT approach has been successfully applied in several studies, e.g. to mesoscale gravity waves (Lu et al., 2005; Koch et al., 2005) and in the characterization and removal of non-stationary and localized vertical structures in atmospheric temperature and density profiles retrieved from satellites (Iannone et al., 2014).

In this work, we used the wavelet software package, developed by Torrence and Compo (1998) in a series of different programming languages (available at http://paos.colorado.edu/research/wavelets/). We applied the CWT method to the AIRWAVE TCWV fields shown in Fig. 4. The analysis was performed in the along track direction (i.e. coincident with the satellite motion) which (in these cases) is nearly aligned with the wind direction and therefore with the wave pattern.

The method computes the energy of spectral components as a function of along track position (associated with a value of latitude, longitude and TCWV amount) and oscillation wavelength. For each along track position, we then selected the wavelength corresponding to the most energetic wave. Repeating the analysis for each vector corresponding to a different across track position, we obtained the results shown in Fig. 6. The left panel shows the analysis for the ATSR-2 orbit number 38086, while the right panel for the co-located AATSR orbit number 2214.

We considered only wave power values within the "95 % confidence interval" which is obtained comparing the spectrum of individual wavelength series against a certain background level (the red noise) (Torrence and Compo, 1998). Furthermore, we selected only wavelength values within the space/frequency region determined using the so-called Cone Of Influence (COI), which is the region where the wavelet power spectra values can be considered unbiased by edge effects. The analysis shows that the majority of the TCWV fluctuations are statistically significant, both for AATSR and for ATSR-2. Wavelengths were detected mainly from 6 to 8 km (associated with azure-green colours in Fig. 6) with amplitudes up to 75 % of the average TCWV amounts. Our findings are confirmed by the theoretical study performed by Nappo (2013). The paper states that the most energetic wavenumber $k_{max}$ can be simply predicted using a mechanical approximation for lee wave development, assuming a spectrum of horizontal wavenumbers with $k_{max} = 4b$, where $b$ is the scale width of the mountain, computed as the distance between the maximum height and the point where the altitude reaches half of the maximum. Since the island of Amorgos is characterized by scale widths ranging from 1 to 2 km, this leads to a prediction of wavelengths ranging from 4 to 8 km, consistent with the wavelet analysis. Our method is analogous to the more rigorous Taylor-Goldstein approach described in (Shutts, 1997), whereby the wind and stability structure in the troposphere determines the resonant wavelength of the trapped wave.

## 5   Conclusions

We have identified lee waves occurrences over the Mediterranean basin from the AIRWAVE total column water vapour dataset. The AIRWAVE data comprises about 20-years of day and night time products over water in clear-sky conditions, derived from the ATSR instrument series. The high ($1 \times 1$ km$^2$) spatial resolution of the dataset is essential for the detection of these

phenomena. We have investigated regions over the Tyrrhenian and Aegean seas, finding that the Greek basin is the place where lee waves are observed with the highest frequency, due to its complex orography and coastlines and to the presence of intense winds and clear-sky conditions during most of the year. Lee waves have been found using an automatic tool that computes the TCWV standard deviation in spatial regions about 200/250 km$^2$ wide and classifies the cases based on the number of clear-sky

measurements and the derived normalised standard deviation.

We verified that the adopted method does not produce "false positive" detections in regions where the presence of lee wave events is not expected (e.g. open ocean). The proposed approach has been intentionally kept simple, however some future improvements could be implemented, such as including in the detection algorithm the caveat of repetitive features in a short horizontal distance along the flow direction. We validated a sub-set of detected lee wave events, finding the presence of similar

structures both in the AIRWAVE dataset and in the correlative data (independent observations or WRF model simulations).

Using the Morlet continuous wavelet transformation analysis, we were able to determine the characteristics of the observed lee waves in a region located South-West of the narrow island of Amorgos, finding wavelengths between 6 to 8 km with amplitudes up to 75 % of the mean TCWV amount. The values are reasonable from a theoretical point of view and were confirmed by a WRF model run performed for the selected case. Therefore, the wavelength analysis approach adopted in this

work has been found to be adequate. Nevertheless, we are investigating the possibility to develop a more sophisticated method for a statistical analysis of these events.

The proposed approach, which does not rely on external information, could in principle be applied to high spatial resolution TCWV products from other satellite instruments. The method is also very efficient from a computational point of view and is therefore suitable for near real-time detection of lee waves from current flying sensors, e.g. the Sea and Land Surface

Temperature Radiometer (SLSTR) and the Ocean and Land Color Instrument (OLCI), both onboard Copernicus Sentinel-3 (Donlon et al., 2012), the two Moderate Resolution Imaging Spectroradiometer (MODIS) instruments onboard Terra and Aqua (Barnes and Salomonson, 1992). The tool could also be applied to other regions of the Earth. Future developments include a possible evaluation of radiative forcing variations associated to wave events and a more accurate characterization of the waves exploiting the "tandem periods" when measurements of two instruments almost overlap (e.g. SLSTR/Sentinel-3a and

SLSTR/Sentinel-3b).

*Author contributions.* EP, EC, SC, BMD conceived the method. MMM performed the WRF model simulations. EP performed the validation, the wavelet analysis and drafted the paper. All authors discussed the results, read and commented on the manuscript.

*Competing interests.* The authors declare that they have no conflict of interest.

*Acknowledgements.* This work has been performed under the ESA/ESRIN IDEAS+ contract no. 4000108531/13/I-NB. MMM gratefully acknowledges the funding from the European Commission (Project "CEASELESS", grant agreement no. 730030). A sincere thank you to Richard Siddans for proofreading the article.

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

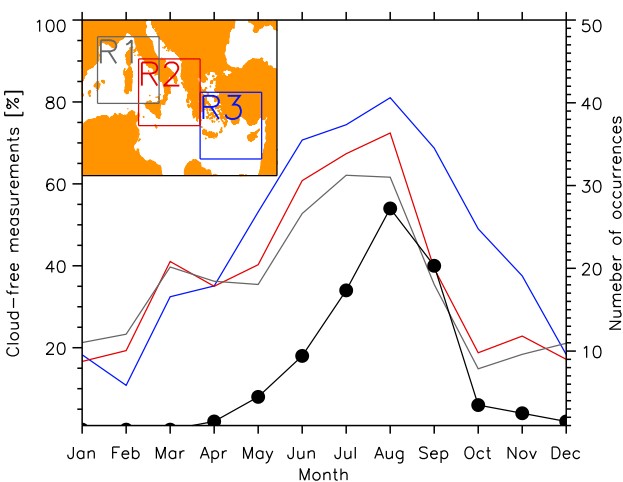

**Figure 1.** Monthly means of the percentage of cloud-free measurements for the three corresponding areas (shown on the top-left map, R1=grey, R2=red, R3=blue) for the year 2003. The total number of lee waves occurrences (ATSR-2 + AATSR for the whole missions) is also reported as a function of the month (black curve, with the scale on the right hand side).

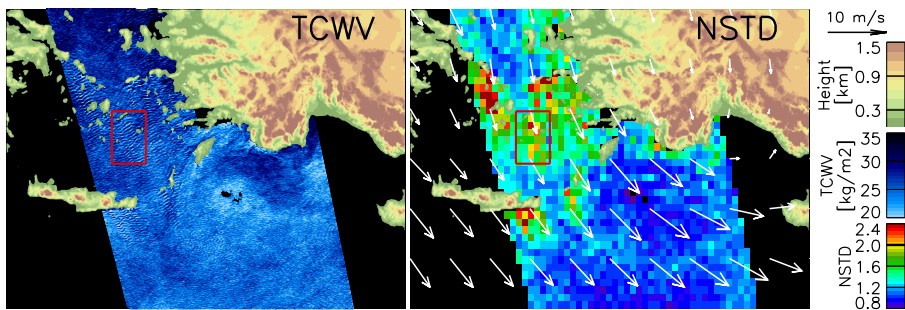

**Figure 2.** Left panel: AIRWAVE TCWV for 2 Aug 2002, AATSR orbit 2214, ascending. Right panel: corresponding normalised standard deviation. The arrows represent the wind speed (length) and direction, as derived from ERA-Interim products at $1.0° \times 1.0°$ resolution. The area delimited by the red box is studied more in detail in Fig. 4.

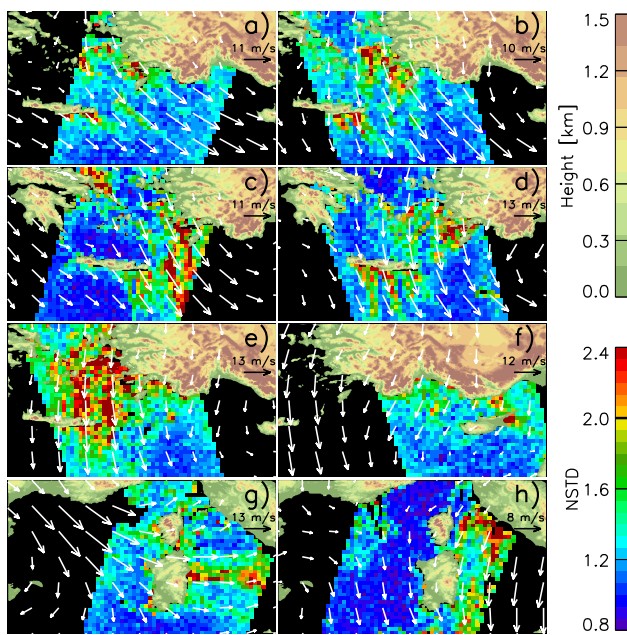

**Figure 3.** Normalised standard deviation for: a) 23 Jul 1997, ATSR-2 orbit 11798, descending; b) 23 Jul 1997, ATSR-2 orbit 11805, ascending; c) 23 Jul 2002, ATSR-2 orbit 37936, descending; d) 26 Apr 2003, AATSR orbit 6036, ascending; e) 15 Jul 2009, AATSR orbit 38558, ascending; f) 19 Jun 2002, ATSR-2 orbit 37456, ascending; g) 2 Aug 2006, AATSR orbit 23121, descending; h) 27 Jul 2002, ATSR-2 orbit 37994, descending. The arrows represent the wind speed (length) and direction, as derived from ERA-Interim products at $1.0° \times 1.0°$ resolution.

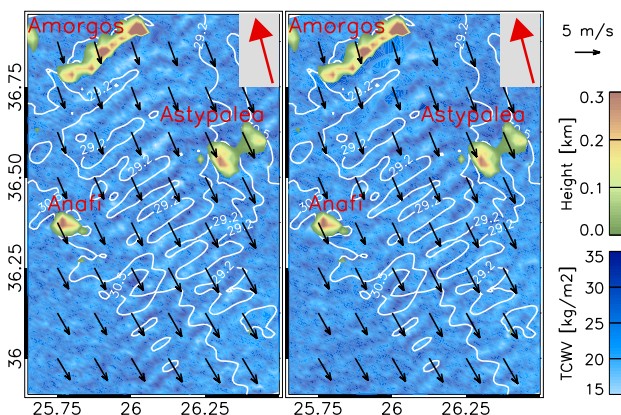

**Figure 4.** AIRWAVE TCWV (bluish colors) for 2 August 2002 over Aegean Sea derived from: ATSR-2, orbit 38086 (ascending), measurements acquired from 20:11:41 to 20:12:00 (left), AATSR, orbit 2214 (ascending), measurements acquired from 19:43:11 to 19:43:30 (right). The black arrows represent wind speed (length) and direction, as derived from ERA-Interim fields interpolated at $0.125° \times 0.125°$ resolution. TCWV as computed from WRF model run (output at 20:00:00) is superimposed as white contour lines for two specific values of 29.2 and 30.5 kg m$^{-2}$. The top-right box shows the satellite flight direction.

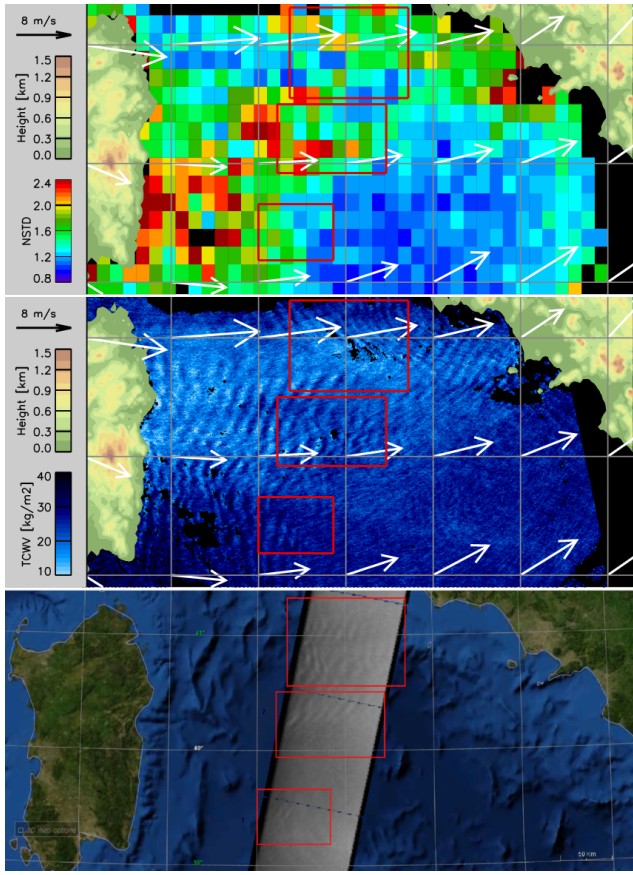

**Figure 5.** 18 July 1997 over Tyrrhenian Sea. Top panel: NSTD derived from the AIRWAVE TCWV fields (shown in the central panel), ATSR-2, orbit 11734, ascending, measurements acquired at 21:14. The arrows represent the wind speed (length) and direction, as derived from ERA-Interim fields at $1.0° \times 1.0°$ resolution. Bottom panel: SAR/ERS-2 single look complex images, track 351, frames 2763, 2781, 2799, 2817, orbit 11728, descending, measurements acquired at 9:58.

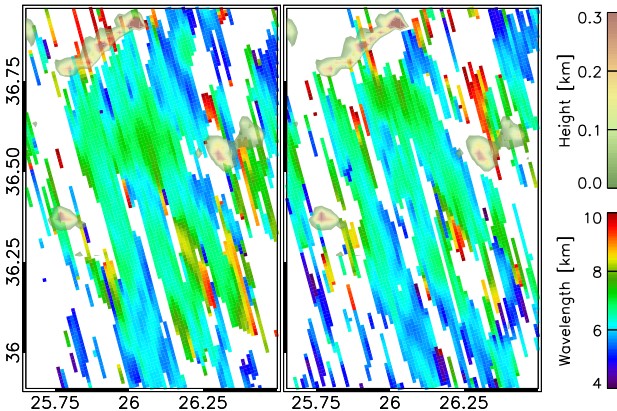

**Figure 6.** Wavelength, indicated by the lower bar with a colour scale ranging from 4 to 10 km, as derived applying the CWT method to the AIRWAVE TCWV products shown in Fig 4. The left panel shows ATSR-2 orbit 38086 (ascending), the right panel AATSR orbit 2214 (ascending). The height of the ground is also shown (upper bar).