# Peer review of "Lee waves detection over the Mediterranean Sea using the Advanced Infra-Red WAter Vapour Estimator (AIRWAVE) Total Column Water Vapor (TCWV) dataset"

_Atmospheric Measurement Techniques, 2019_

## Referee Comment (RC1) · Anonymous Referee #1 · 9 May 2019

The authors have addressed the issue of detecting lee waves from satellite observations over the Mediterranean basin. Although the issue is not new, to my knowledge this is the first time the analysis has been performed using data from ATSR instrument series. Lee waves are normally detected in cloudy skies using clouds themselves as tracers. In the present paper, the tracer is total column water vapour derived from an algorithm developed from the same authors. Therefore, they focus on clear sky, which demands for a robust and accurate TCWV estimate. In this respect, the results also provide an indirect validation of the retrieval algorithm. The authors have

provided a very exhaustive and comprehensive analysis of the results, which encompasses comparison with other satellite instruments and simulations from a numerical weather prediction model (WRF). The effort of authors also includes an automated scheme to search for clear sky and favourable lee wave conditions, and a final analysis for the estimate of the wave parameters, such as amplitude, wavelength and phase.

I have to say that I enjoyed reading the paper and therefore I have not so many questions to ask. Maybe one point they could clarify is that lee waves are generally stationary waves, which enables one to recover the wave-parameters from 2-D spatial patterns as seen from the satellite, rather than 3-D. Also, they could clarify how important is the hypothesis of stationarity for their method and analysis to work and what are the limitations, if any.

---

## Referee Comment (RC2) · Anonymous Referee #2 · 16 May 2019

Review of "Lee waves detection over the Mediterranean Sea using the Advanced Infra-Red WAter Vapour Estimator (AIRWAVE) Total Column Water Vapor (TCWV) dataset" by E. Papandrea et al., 2019

The authors describe the use of total column water vapour (TCWV) data from ATSR-2 and AATSR satellite radiometer measurements to study lee waves, which create a phase imprint on the former quantity resulting from the lee wave perturbations of atmospheric motion. The TCWV retrieval algorithm was created by a number of the same authors. They also here develop a gridded lee wave diagnostic tool based on

the variability of TCWV within a grid square, which is shown to be largely instrument-independent due to its focus on relative variability compared to areas measured by the same instrument where lee waves are weak. After applying to 20 years of satellite data, a number of case studies are extracted to show the diagnostic's representativity of lee wave activity. Aside from the usefulness of the authors' method for general lee wave research, as model resolution increases, resolution of lee waves by regional scale numerical weather models is becoming more commonplace, and it is important that the models represent these phenomena accurately due to their impacts on surface winds and variability, as well as orographic drag on the atmosphere. To validate this properly, distributed measurement methods are required, to which common observation networks are not well suited. A number of methods including visible satellite imagery, focussed field campaigns and spaceborne synthetic aperture radar wind measurements can be used but each has its weakness, and the more corroborating sources of information are available the better. This paper provides a good demonstration of another rich source of information which can be used in the above capacity, or simply as a real-time detection method for meteorological guidance providers, and which builds upon examples of similar data exploitation in the literature. It is well written and straightforward to understand. Subject to addressing a number of minor queries and questions below ("p" referring to page and "l." referring to line numbers), I recommend it for publication in Atmospheric Measurement Techniques.

p1:

l.11. Redundant full-stop before citations.

l.13. M. Teixiera et al. have recently published a number of papers studying drag due to trapped lee waves, which could be referenced here (DOI:10.1002/qj.2008, 10.1175/JAS-D-12-0350.1, 10.1175/JAS-D-16-0199.1, 10.1002/qj.3177).

p2:

l.25-34. Can the authors outline the basic mechanism by which lee waves cause TCWV

variation?

p4:

l.5-6. Does significant variation of the baseline TCWV occur in different cases depending on the moistness of the atmosphere or other factors, and does this in turn influence the value of NTSD for a given wave amplitude (as defined in terms of vertical velocity or some other more direct parameter)? Alternatively, do other factors (such as the propensity for sea surface evaporation due to whatever cause) influence the value of TCWV for a given wave amplitude?

l.10. Can the authors give some idea of the areas and criteria used?

l.17. By what criteria was "best performance" determined?

l.23-24. Over what interval is TR1 calculated (or how many scenes)? Or do the authors mean "cloud-free grid cells" and not "cloud-free scenes" on line 23? Can the authors comment on the possible effects of variable coverage of TCWV due to missing data on the values of TR1 and TR2? For instance, if data is missing in one case close in the lee of a mountainous island, and in another case the opposite.

p5:

l.6-7. The correlation is reasonable in an absolute sense, but the fall to zero in winter despite a still-substantial minority of cloud-free scenes is notable - can the authors speculate on reasons for this? For instance, are significant lee waves in winter ubiquitously accompanied by cloud (due to the synoptic conditions in which they arise)? Or some other reason?

l.22. Why does wavelength influence the NTSD? Assuming, for simplicity, sinusoidal behaviour, I would think the NTSD should be the same for a given amplitude regardless of wavelength. As the wavelength approaches the data resolution, this could emphasise extreme TCWV values, increasing NTSD, but will also smooth them, with a compensating (or greater) diminution of the NTSD value.

Figures 2-5. It would be useful to have a key for the ERA-Interim wind vectors shown on these plots, indicating the wind speed for a given vector size (e.g. 20 m/s).

p6:

l.9. There is some redundancy/repetition within this sentence.

p7:

l.33. This is helpful but the link is deprecated - at the moment it provides information and links through to https://esar-ds.eo.esa.int/oads/access/, the new catalogue. It would be useful to add the latter link in case the former legacy page becomes defunct.

Figure 5. There appear to be some large NTSD values in grid squares to the left of the lowermost red box in the top panel, an area with a high degree of missing data, although the TCWV variability in the same area seems rather tame - is this some artefact of missing data? The SAR and TCWV data patterns (i.e. wave phase) are not quantitatively alike in the respective panels, presumably due to a time offset. Rather than quoting orbits for these data, which mean little to the reader, could the authors state the overpass times corresponding to the SAR and TCWV images.

Figure 6. "Wavelength" is spelled wrong in the axis title.

p9:

l.2-7. It would be useful also to compute at least a rough trapped wave wavelength based on the Scorer parameter (or very crudely, 2*Pi*U/N over some layer close to the mountian top) for comparison with these values.

l.17. I assume the method may struggle to distinguish lee waves from e.g. the Bora outflows common in the Adriatic, convective storms or other features that could create large NTSD values? I don't wish to invalidate the authors' method, which has virtues of simplicity and transparency, is relatively instrument-independent, and is effective where we know lee waves are the dominant cause of TCWV variance. I do, however,

wonder if there are certain caveats, or if a further layer of analysis or diagnosis would be required before the algorithm could be reliably operated in a fully automated way.

l.28-29. Given the potential usefulness of this technique (and the low frequency of overpasses), are the authors able to highlight a more comprehensive list of instruments for which this type of analysis would possible?

Final comment (an inversion of comments above concerning p4. l5-6.): Have the authors done any study concerning the vertical velocities corresponding to a given NSTD value, for instance a correlation plot at the level for which lee waves most strongly influence TCWV? Although the correlation may contain some scatter, the equivalence would presumably be less crude than, for instance, the commonly-used inference from satellite images in the visible range. If not, this would be a useful future study, which could be carried out using either model data or, where vertical velocity data may be obtained, observations.
* * *

---

## Author Comment (AC1) · 1 Jul 2019

The authors have addressed the issue of detecting lee waves from satellite observations over the Mediterranean basin. Although the issue is not new, to my knowledge this is the first time the analysis has been performed using data from ATSR instrument series. Lee waves are normally detected in cloudy skies using clouds themselves as tracers. In the present paper, the tracer is total column water vapour derived from an algorithm developed from the same authors. Therefore, they focus on clear sky, which demands for a robust and accurate TCWV estimate. In this respect, the results also provide an indirect validation of the retrieval algorithm. The authors have provided a very exhaustive and comprehensive analysis of the results, which encompasses comparison with other satellite instruments and simulations from a numerical weather prediction model (WRF). The effort of authors also includes an automated scheme to search for clear sky and favourable lee wave conditions, and a final analysis for the estimate of the wave parameters, such as amplitude, wavelength and phase.
I have to say that I enjoyed reading the paper and therefore I have not so many questions to ask. Maybe one point they could clarify is that lee waves are generally stationary waves, which enables one to recover the wave-parameters from 2-D spatial patterns as seen from the satellite, rather than 3-D. Also, they could clarify how important is the hypothesis of stationarity for their method and analysis to work and what are the limitations, if any.

We inserted in the text at **p8** l.12 the following sentences:
"Lee waves are stationary waves; this property enables to recover the wave parameters from the 2-D spatial patterns seen from the satellite. Small variations in the lee waves induced features may occur, but their time scale is much longer than the short time required for the coverage of the considered areas, so that they should not affect our results".

---

## Author Comment (AC2) · 1 Jul 2019

Review of "Lee waves detection over the Mediterranean Sea using the Advanced Infra-Red WAter Vapour Estimator (AIRWAVE) Total Column Water Vapor (TCWV) dataset" by E. Papandrea et al., 2019
The authors describe the use of total column water vapour (TCWV) data from ATSR-2 and AATSR satellite radiometer measurements to study lee waves, which create a phase imprint on the former quantity resulting from the lee wave perturbations of atmospheric motion. The TCWV retrieval algorithm was created by a number of the same authors. They also here develop a gridded lee wave diagnostic tool based on the variability of TCWV within a grid square, which is shown to be largely instrument-independent due to its focus on relative variability compared to areas measured by the same instrument where lee waves are weak. After applying to 20 years of satellite data, a number of case studies are extracted to show the diagnostic's representativity of lee wave activity. Aside from the usefulness of the authors' method for general lee wave research, as model resolution increases, resolution of lee waves by regional scale numerical weather models is becoming more commonplace, and it is important that the models represent these phenomena accurately due to their impacts on surface winds and variability, as well as orographic drag on the atmosphere. To validate this properly, distributed measurement methods are required, to which common observation networks are not well suited. A number of methods including visible satellite imagery, focussed field campaigns and spaceborne synthetic aperture radar wind measurements can be used but each has its weakness, and the more corroborating sources of information are available the better. This paper provides a good demonstration of another rich source of information which can be used in the above capacity, or simply as a real-time detection method for meteorological guidance providers, and which builds upon examples of similar data exploitation in the literature. It is well written and straightforward to understand. Subject to addressing a number of minor queries and questions below ("p" referring to page and "l." referring to line numbers), I recommend it for publication in Atmospheric Measurement Techniques.

**p1:**
l.11. Redundant full-stop before citations.
We suppose that this comment is referred to page 2 instead of page 1.
Done.

l.13. M. Teixiera et al. have recently published a number of papers studying drag due to trapped lee waves, which could be referenced here (DOI:10.1002/qj.2008, 10.1175/JAS-D-12-0350.1, 10.1175/JAS-D-16-0199.1, 10.1002/qj.3177).
We suppose that this comment is referred to page 2 instead of page 1.
We included all the above-mentioned suggested papers.

**p2:**

l.25-34. Can the authors outline the basic mechanism by which lee waves cause TCWV variation?

In order to clarify the basic mechanism leading to the TCWV modulation caused by lee waves, we inserted in the text (**p2** l.25) the following sentences:
"Lee waves produce oscillations in different atmospheric variables, including vertical velocity. As a consequence, the air is alternately lifted and lowered, causing convergence and divergence of air in the warmer moist bottom layer, which is seen as bands in the TCWV field (Lyapustin et al., 2014)."

Lyapustin, A., Alexander, M. J., Ott, L., Molod, A., Holben, B., Susskind, J., and Wang, Y. (2014), Observation of mountain lee waves with MODIS NIR column water vapor, *Geophys. Res. Lett.*, **41**, 710– 716, doi:10.1002/2013GL058770.

**p4:**
l.5-6. Does significant variation of the baseline TCWV occur in different cases depending on the moistness of the atmosphere or other factors, and does this in turn influence the value of NTSD for a given wave amplitude (as defined in terms of vertical velocity or some other more direct parameter)? Alternatively, do other factors (such as the propensity for sea surface evaporation due to whatever cause) influence the value of TCWV for a given wave amplitude?

The following paragraph has been added to the manuscript, **p4** l.14.
"Based on ERA-Interim reanalysis, Wypych et al. (2018) provided an analysis of temporal and spatial variability of TCWV for Europe. Differences strongly depend on air temperature and on latitude; other determinants include local factors, such as the presence of water or land.
Atmospheric circulation is a key factor for the moisture content in winter. In contrast, evaporation from the sea provides a relevant source of moisture in Mediterranean areas especially in autumn, when the air temperature is still high and the air is able to absorb water vapor emitted by the heated sea surface.
About the year-to-year variability, a fairly even distribution of TCWV is characteristic of the Mediterranean Sea, with changes in TCWV less than 4.5 kg m$^{-2}$. The monthly variability is also relatively small in the Mediterranean, being less than 20 % of the monthly mean TCWV in summer. For these reasons, the influence of the variation of the baseline TCWV on the value of NTSD is limited for our purposes."

Wypych, A.; Bochenek, B.; Różycki, M. Atmospheric Moisture Content over Europe and the Northern Atlantic. *Atmosphere* **2018**, *9*, 18.

l.10. Can the authors give some idea of the areas and criteria used?
The choice of the area and of the criteria was mostly driven by the purpose to homogenise the STD differences caused by the different noise in the measured BTs among the ATSR instrument series. The seasons have a minor impact on the observed background STD values. In order to give more details about the criteria used, we inserted in the text at **p4** l.10 the following sentence:
"For this purpose, the identified areas have been selected on the bases of wind speed and direction (< 2 m s$^{-1}$ and not downwind to land)."

l.17. By what criteria was "best performance" determined?
We tested, in particular, three grid sizes, 0.05°x0.05°, 0.15°x0.15°, and 0.25°x0.25°. All the three grids were found to allow for lee waves to be detected. The 0.25°x0.25° was found too coarse to take into account the small-scale spatial variability of the waves at the edge of the areas where these phenomena occur and also too coarse to capture the geographic details of the complex coastline structure in the considered region. The 0.05°x0.05° grid was discarded

for robustness considerations: the relatively low number of elements which fall inside may indeed cause too weak reduction of the associated noise in the average procedure.
At the end of the sentence at **p4** l.15-18, we added the following text "…in terms of both the ability to capture coastline details and the robustness of statistics".

l.23-24. Over what interval is TR1 calculated (or how many scenes)? Or do the authors mean "cloud-free grid cells" and not "cloud-free scenes" on line 23?
The referee is right, we mean "cloud-free grid cells" instead of "cloud-free scenes". We changed the text in the paper.
Can the authors comment on the possible effects of variable coverage of TCWV due to missing data on the values of TR1 and TR2? For instance, if data is missing in one case close in the lee of a mountainous island, and in another case the opposite.
The first test (TR1) is related to the number of grid cells having cloud-free measurements. This number, reflecting only the percentage of available (cloud-free) grid cells with respect to the total number of grid cells within the scene, is not significantly affected by the presence of waves in the TCWV values.
The second test (TR2) is only marginally affected by missing data. The only concern is that missing data may generate a less robust statistics. However, if cloudy measurements are correctly detected by the level 1 cloud mask, this is not a major issue.
If the number of cloud-free (or available) measurements within the grid cell is less than 20%, the cell is not considered. This also enables to avoid grid cells having only sporadic measurements, due e.g. to the presence of land within the considered cell. Please see also below the reply to Figure 5 comment on "artefact of missing data".

**p5:**
l.6-7. The correlation is reasonable in an absolute sense, but the fall to zero in winter despite a still-substantial minority of cloud-free scenes is notable - can the authors speculate on reasons for this? For instance, are significant lee waves in winter ubiquitously accompanied by cloud (due to the synoptic conditions in which they arise)? Or some other reason?
The zero selection in winter may, as the reviewer correctly suggests, be caused by the fact that winter lee waves are frequently accompanied by cloud-conditions. Another reason can be attributed to statistics: what is shown in the plot is the % of cloud-free measurements; when the number of cloud-free measurements is low (20-30%), the number of grid cells having cloud-free measurements > 20% is also low and is therefore unlikely that the "scene" could pass the criterium TR1 (number of cloud-free grid cells > 70%).
We reckon that the first explanation would add value to the paper, therefore we included the following sentence at **p5** l.7:
"One reason for no occurrence in the winter months could be attributed to the frequent presence of lee waves accompanied by cloudy conditions."

l.22. Why does wavelength influence the NTSD? Assuming, for simplicity, sinusoidal behaviour, I would think the NTSD should be the same for a given amplitude regardless of wavelength. As the wavelength approaches the data resolution, this could emphasise extreme TCWV values, increasing NTSD, but will also smooth them, with a compensating (or greater) diminution of the NTSD value.
The referee is right and Figure a has been added (only in this document) in order to clarify this point. If the grid cell is at least large wavelength/2 (black, blue, green curves) the STD (or NSTD) is almost independent of the wavelength, while the STD is lower for the red curve, where the wavelength/2 is larger than the grid cell size.

We removed in the revised text "shorter wavelengths or/and by".

[Figure]

Figure a: sinusoidal curves characterised by different wavelengths (with black, blue, green and red colours). The corresponding STD, computed for the abscissa range shown in the figure, is shown on the top of the figure using the same colours.

Figures 2-5. It would be useful to have a key for the ERA-Interim wind vectors shown on these plots, indicating the wind speed for a given vector size (e.g. 20 m/s).
The figures have been modified. We used thickness of the vectors to indicate wind speed and we also added a legend. We made this choice to offer to the readers a more accurate evaluation of the wind magnitudes.

**p6:**
l.9. There is some redundancy/repetition within this sentence.
We modified this sentence:
"This lee wave detection was then compared to the outputs of the WRF numerical limited area model (see Sect 3.1)."

**p7:**
l.33. This is helpful but the link is deprecated - at the moment it provides information and links through to https://esar-ds.eo.esa.int/oads/access/, the new catalogue. It would be useful to add the latter link in case the former legacy page becomes defunct.
We added the new link and we replaced "allows" with "allowed".
"The picture has been obtained using the Earth Observation Link (EOLi, https://earth.esa.int/web/guest/eoli) European Space Agency's client for Earth Observation Catalogue Service. The EOLi tool allows allowed the selection of Earth Observation products acquired by the ERS and ENVISAT satellites and the display of the related images on the top of an orthographic representation of the Earth. The service has been recently replaced by the ESA Simple Online Catalogue (https://esar-ds.eo.esa.int/oads/access/)."

Figure 5. There appear to be some large NTSD values in grid squares to the left of the lowermost red box in the top panel, an area with a high degree of missing data, although the TCWV variability in the same area seems rather tame - is this some artefact of missing data?
The referee is right saying that the NSTD values is larger than expected and also larger with respect to the surrounding boxes. However, in our method the NSTD is not considered if the

number of available measurements is less than 20%, and we can see that two grid boxes are not shown to the left of the lowermost red box. Therefore, the enhancement in this particular area may be due to two causes: a relatively large number of missing data and some erroneous measurements, e.g. affected by thin clouds not filtered by the cloud mask.
We inserted in the text the following sentence at **p7** l.31:
"Once again, the NSTD enhancements are largely correlated with the lee wave patterns in the TCWV fields. However, in the presence of missing data due to clouds, it may happen that the NSTD values increase more than expected. The reason for this behaviour can be ascribed, for example, to measurements affected by thin clouds not filtered out by the ATSR cloud mask, thus affecting the TCWV retrieval. A secondary cause may be the presence of clouds, reducing the number of elements that can be used within the grid cells."
The SAR and TCWV data patterns (i.e. wave phase) are not
quantitatively alike in the respective panels, presumably due to a time offset. Rather
than quoting orbits for these data, which mean little to the reader, could the authors
state the overpass times corresponding to the SAR and TCWV images.
Inserted time overpasses in the caption of the figure.

Figure 6. "Wavelength" is spelled wrong in the axis title.
Corrected

**p9:**
l.2-7. It would be useful also to compute at least a rough trapped wave wavelength
based on the Scorer parameter (or very crudely, 2*Pi*U/N over some layer close to the
mountian top) for comparison with these values.
A rough estimate of the trapped-wave wavelength provided by 2*Pi*U/N is shown hereafter at 20:00 UTC, 2 August 2002. U is calculated at 900 hPa, as representative wind speed in the lower levels (the wind impinges almost perpendicular to Amorgos island), while N is calculated in the layer 850-1000 hPa. Although with some difference, the model and the observations show similarly the maximum wavelengths (of about 8 km) in the wake of the Amorgos island.

[Figure]

l.17. I assume the method may struggle to distinguish lee waves from e.g. the Bora
outflows common in the Adriatic, convective storms or other features that could create

large NTSD values? I don't wish to invalidate the authors' method, which has virtues of simplicity and transparency, is relatively instrument-independent, and is effective where we know lee waves are the dominant cause of TCWV variance. I do, however, wonder if there are certain caveats, or if a further layer of analysis or diagnosis would be required before the algorithm could be reliably operated in a fully automated way.

It is true that other mesoscale features, such as Bora or Mistral flows and thunderstorm outflows may create large NTSD values. However, the proposed methodology was designed as a first approach to the problem. Improvements, such as including in the algorithm the caveat of repetitive features in a short horizontal distance along the flow direction, or analyzing other data to integrate those from the ATSR instruments, are planned for the future.

l.28-29. Given the potential usefulness of this technique (and the low frequency of overpasses), are the authors able to highlight a more comprehensive list of instruments for which this type of analysis would possible?

We added at the list: MODIS/Terra, MODIS/Aqua, OLCI. The sentence was modified: "…e.g. the Sea and Land Surface Temperature Radiometer (SLSTR) and the Ocean and Land Color Instrument (OLCI), both onboard Copernicus Sentinel-3 (Donlon et al., 2012), the two Moderate Resolution Imaging Spectroradiometer (MODIS) instruments onboard Terra and Aqua (Barnes and Salomon, 1992)."

Final comment (an inversion of comments above concerning p4. l5-6.): Have the authors done any study concerning the vertical velocities corresponding to a given NSTD value, for instance a correlation plot at the level for which lee waves most strongly influence TCWV? Although the correlation may contain some scatter, the equivalence would presumably be less crude than, for instance, the commonly-used inference from satellite images in the visible range. If not, this would be a useful future study, which could be carried out using either model data or, where vertical velocity data may be obtained, observations.

This is a very interesting point that we will take into consideration for a future study.

We substituted cloud free with "cloud-free" **p4** l.17 for uniformity

---

## Author Response (AR2)

*The authors have addressed the concerns raised in my original review. I have a few follow-up comments, which I count as representing minor revisions, subject to which I recommend acceptance of the paper.*
*Page and line numbers below refer to the original manuscript, to match the original review comments and authors' response. Two other new comments are added at the end, referring to the latest manuscript.*

*p2 l25-34. So is this a deepening of the moist layer?*
The underlined text was added in the revised manuscript.
"*Lee waves produce a deepening of the moist layer and oscillations in different atmospheric variables, including vertical velocity.*"

*p5 l6-7. For clarity I'd insert "TCWV wave" between "no" and "occurrence" in this added sentence.*
Inserted in the revised text.

*p5 l22. There are two relevant scale limits here, the grid size for aggregation, and the satellite resolution. As wavelength decreases below 8 satellite pixels, so that wave effects are increasingly not resolved, NTSD will decrease. As longer wavelengths approach the aggregation box used to calculate NTSD, NTSD will again decrease since only part of the wave is captured in the box (although it is not certain to me that this is occurring in Figure 2 as the authors suggest; wavelengths seem generally less than the aggregation box size). A simple statement that "shorter wavelengths ... generate larger standard deviations" is therefore not adequate. I think the authors just need to make a clear statement about these relevant scales and how they would affect the NTSD data. Of course if the aggregation box size is in fact impacting the NTSD value, it begs the question in the mind of the reader as to why a larger, or perhaps adaptive, grid box size was not used (although I appreciate that the method used is presented as a prototype).*
We replaced the sentence "shorter wavelengths ... generate larger standard deviations" with "*The used approach, which makes use for simplicity of a regular aggregation box size used to calculate the NSTD, works almost uniformly for the majority of typical wavelengths. Nevertheless, small effects may arise both in case of wavelengths lower than half of the box size (wave effects are not completely resolved) or in case of wavelengths approaching the box size (only part of the wave is captured within the box), resulting in a decreasing of the NSTD value.*"

*Figures 2-5. I'm not really sure how using the vector thickness, which is hard to discern and only allows thresholds of 3, 6 and 9 to be perceived (with difficulty), is better than using the*

*vector length, which would be obvious and is common practice (one example of, say, 6 or 9 m/s would suffice).*

*We followed the suggestion of the reviewer, replacing the vector thickness with the more widely used vector length.*

*p9. l2-7. Have the authors considered making some comment in the manuscript regarding this? The exercise represents a crude analogue to a more rigorous Taylor-Goldstein approach (see e.g. Shutts, 1997, https://doi.org/10.1017/S1350482797000340), whereby the wind and stability structure in the troposphere decides the resonant wavelength of the trapped wave. A vertical structure consideration is thus very relevant to the trapped wave wavelength likely to occur, while the orographic forcing length scale relates more to the amplitude.*

*We inserted at the end of Section 4, the following sentence "Our method is analogous to the more rigorous Taylor-Goldstein approach described in Shutts (1997), whereby the wind and stability structure in the troposphere determines the resonant wavelength of the trapped wave."*

*p9. l17. It feels worth mentioning this future work idea in the manuscript, perhaps in relation to the sentence starting 'We verified that the adopted method do not produce "false positive" detections...'?*

*We included, after the sentence proposed by the reviewer "The proposed approach has been intentionally kept simple, however some future improvements could be implemented, such as including in the detection algorithm the caveat of repetitive features in a short horizontal distance along the flow direction."*

*New comments on latest manuscript:*

*p6 l13-15. Surely it is waves that propagate in the wave duct, not "shear"? "Wakes" also are controlled in a different way.*

[revised manuscript text omitted]